# Network Signatures from Image Representation of Adjacency Matrices: Deep/Transfer Learning for Subgraph Classification

## Abstract

We propose a novel subgraph image representation for classification of network fragments with the target being their parent networks. The graph image representation is based on 2D image embeddings of adjacency matrices. We use this image representation in two modes. First, as the input to a machine learning algorithm. Second, as the input to a pure transfer learner. Our conclusions from multiple datasets are that

- deep learning using structured image features performs the best compared to graph kernel and classical features based methods; and,
- pure transfer learning works effectively with minimum interference from the user and is robust against small data.

## 1 Introduction

With the advent of big data, graphical representation of information has gained popularity. Being able to classify graphs has applications in many domains. We ask, "Given a small piece of a parent network, is it possible to identify the nature of the parent network (Figure 1)?" We address this problem using structured image representations of graphs.

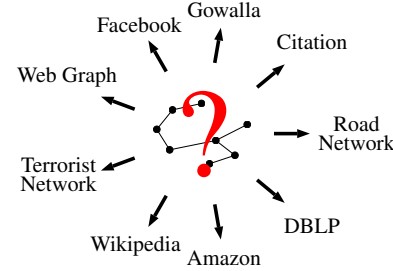

Figure 1: The Problem

Adjacency matrices are notoriously bad for machine learning. It is easy to see why, from the unstructured image of a small fragment of a road network, in figure (a) below. Though the road network is structured, the random image would convey little or no information to machine learning algorithms (in the image, a black pixel at position $(i, j)$ corresponds to an edge between nodes $i$ and $j$). Reordering the vertices (figure (b) below) gives a much more structured image *for the same subgraph as in (a)*. Now, the potential to learn distinguishing properties of the subgraph is evident. We propose to exploit this very observation to solve a basic graph problem (see Figure 1). The datasets mentioned in Figure 1 are discussed in Section 2.4.

(a)
Unstructured image
of adjacency matrix
(road network)

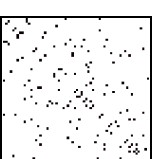

(b)
Structured image of
adjacency matrix
(road network)

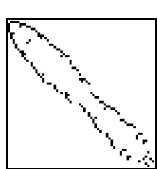

We stress that both images are *lossless* representations of the *same* adjacency matrix. We use the structured image to classify subgraphs in two modes: (i) Deep learning models on the structured image representation as input. (ii) The structured image representation is used as input to a transfer learner (Caffe: see Section 2.3) in a pure transfer learning setting without any change to the Caffe algorithm. Caffe outputs top-$k$ categories that best describe the image. For real world images, these Caffe-descriptions are human friendly as seen in Figure 2a. However, for network-images, Caffe

| Classification | Score |
| --- | --- |
| Pug | 0.75 |
| Bull Mastiff | 0.13 |
| Brabancon Giffon | 0.04 |
| French Bulldog | 0.02 |
| Muzzle | 0.01 |

| Classification | Score |
| --- | --- |
| Window Screen | 0.29 |
| Digital Clock | 0.07 |
| Window Shade | 0.06 |
| Scoreboard | 0.05 |
| Oscilloscope | 0.04 |

(a) An image of a dog

(b) Structured image of Facebook subgraph

Figure 2: An image of a dog and a structured image of a Facebook graph sample vs their corresponding maximally specific classification vectors returned by Caffe

gives a description which doesn't really have intuitive meaning (Figure 2b). We map the Caffe-descriptions to vectors. This allows us to compute similarity between network images using the similarity between Caffe description-vectors (see Section 2).

## 1.1 OUR CONTRIBUTIONS

The significant difference between our work and previous approaches is that we transform graph classification into image classification. We propose an image representation for the adjacency matrix. We use this representation as input to machine learning algorithms for graph classification, yielding top performance. We further show that this representation is powerful enough to serve as input to a pure transfer learner that has been trained in a *completely unrelated image domain*.

**The Adjacency Matrix Image Representation.** Given a sample subgraph from a parent network, the first step is to construct the image representation. We illustrate the workflow below.

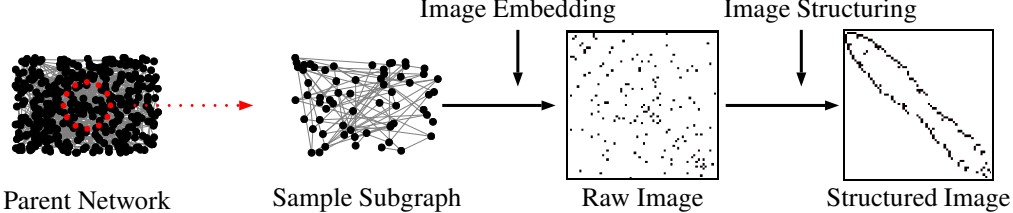

We use the novel method proposed in Wu et al. (2016) which produces an adjacency matrix that is invariant to permutations of the vertices in the adjacency matrix. The image is simply a "picture" of this permutation-invariant adjacency matrix.

**Deep Learning Using the Adjacency Matrix Image Representation.** We train deep image classifiers (discussed in Section 2.2) on our image representation as in Figure 3.

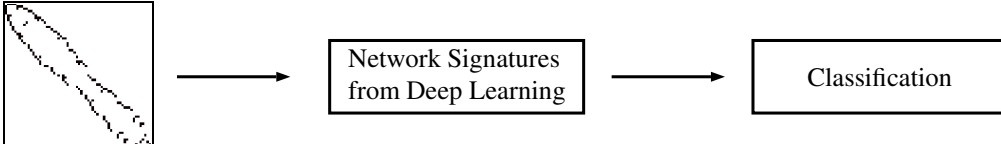

Figure 3: Classification of structured image embeddings using Deep Learning

We compared performance with several methods, including graph kernel classifiers and classifiers based on standard topological features of the graph. Our image representation performs best.

**Transfer Learning Using the Adjacency Matrix Image Representation.** When data is scarce or there are many missing labels, a popular option is transfer learning to leverage knowledge from some other domain. Typically the other domain is closely related to the target application. It is unusual for learning in a completely unrelated domain to be transferable to a new target domain. We

show that our image representation is powerful enough that one can *directly transfer learn* from the real world image domain to the network domain (two completely unrelated domains). That is, our image representation provides a link between these two domains enabling classification in the graph domain to leverage the wealth of techniques available to the image domain.

The image domain has mature pre-trained models based on massive data. For example, the open-source Caffe deep learning framework is a convolutional neural network trained on the ImageNet data which can recognize everyday objects like chairs, cats, dogs etc. (Krizhevsky et al. (2012)). We use Caffe *as is*. Caffe is a black box that provides a distribution over image classes which we refer to as the Caffe-classification vector. The Caffe results are then mapped back into the source domain using a distance-based heuristic e.g. Jaccard distance and $K$-nearest neighbors as in Figure 4.

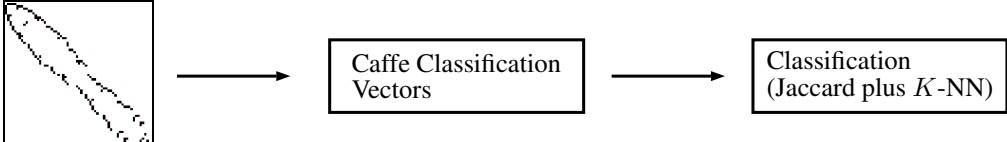

Figure 4: Classification of structured images using the classification vectors obtained from Caffe

Images, *not graphs*, are passed through the Caffe deep neural network, and as we shall show, one can get good performance from as little as 10% of the training data used in the *ab initio* machine learning approach. It is quite stunning that such little training data together with un-tweaked transfer learning from a completely unrelated domain can perform so well. The reason is that our image representation provides very structured images (human-recognizable) for real world networks. Though these images are not traditional images like those used in training Caffe, Caffe still maps the different structured images to different distributions over its known classes, hence we are able to *transfer* this knowledge from Caffe to graph classification.

## 1.2 Related Work

Wu et al. (2016) introduced the problem we study: Can one identify the parent network from a small subgraph? How much does local information reveal about the parent graph at the global level? We approach the problem from the supervised setting and the unsupervised transfer learning setting.

There is previous work on similar problems using graph kernels (Kashima & Inokuchi (2002); Kashima et al. (2003); Kudo et al. (2004)). Such methods use kernels to compute similarities between graphs and then algorithms like SVM for classification. Choosing kernels is not straightforward and is certainly not a *one-size-fits-all* process. Further, these kernel methods do not scale well for very large graphs. We compare with one such method proposed by Shervashidze et al. (2011).

Another approach is to constuct feature vectors from topological attributes of the subgraph (Li et al. (2011)). The topological characteristics of social networks have been extensively studied by Ahn et al. (2007); Ferrara & Fiumara (2012); Sridharan (2011). The shortcomings of this approach are that it is difficult to come up with a *master set* of features that can be used to represent graphs from different domains. For example, assortativity could be an important feature in social networks while being of little significance in case of road networks. It is hard to identify beforehand what features need to be computed for a given problem, thus leading to a trial and error scenario. One of the methods we compare is logistic regression.

Transfer learning is useful when a classification task in one domain can leverage knowledge learned in a related domain (see Pan & Yang (2010) for an extensive survey). Raina et al. (2007) introduced a method called *self-taught learning* which takes advantage of irrelevant unlabeled data to boost performance. Zhu et al. (2011) discuss heterogeneous transfer learning where they use information from text data to improve image classification performance. Quattoni et al. (2008) create a new representation from kernel distances to large unlabeled data points before performing image classification using a small subset of reference prototypes.

The rest of the paper is organized as follows. In Section 2 we give more details of our approaches to subgraph classification. Section 3 present the results comparing the performance of our approach with other approaches. We conclude in Section 4 with some possible future directions.

## 2 METHODOLOGY

Given fragments of a large network we first obtain the structured image representation. In the transfer setting, we pass these training samples to the classifiers and record the results. In the unsupervised setting, we feed the training samples to Caffe framework and obtain the label-vectors. To classify a test subgraph, we first obtain its label-vector through Caffe, compute the distance between the test vector and the training vectors, and classify using majority class of the nearest$-k$ vectors. We explain each step in detail below.

### 2.1 IMAGE EMBEDDINGS OF GRAPHS

An adjacency matrix of a graph can be thought of as a monochrome image with 1s corresponding to dark pixels and 0s corresponding to white pixels. This observation, however, is of limited practical use since it is not permutation invariant. We utilize a novel technique for producing a permutation-invariant ordering of the adjacency matrix, first given in Wu et al. (2016). The authors describe several ways to sort an adjacency matrix like page rank, degree based sorting, etc. They show that the BFS-like approach introduced works best. The ordering starts with the node of highest degree (ties are broken using $k$-neighborhood size for $k = 2$, then $k = 3, \dots$). Subsequently, the ordering proceeds based on a combination of shortest paths and degrees. Details can be found in Wu et al. (2016). The ordering scheme results in permutation-invariant adjacency matrices from which we obtain structured images.

We observe that the image embeddings have enough structure so that even the human eye can distinguish between different networks without much effort. This is the intuition behind our approach. Neural networks are highly successful in recognizing real world objects that have high structural properties. For example, all dogs have similar features although they are individually different.

### 2.2 DEEP LEARNING FOR EXTRACTING NETWORK SIGNATURES

Our primary objective is to use deep learning to learn to classify subgraphs to their parent networks using our image representation of the subgraph. However, we also tested a wide variety of other methods as outlined in the table below (more details are in the Appendix, including the references).

| ID | Method |
|---|---|
| DBN | Deep Belief Network: layers of Restricted Boltzmann Machines |
| CNN | Convolutional Neural Network: deep, feed-forward neural networks |
| SdA | Stacked denoising Auto-Encoder: NN layers trained as auto-encoders |
| DCNN | Diffusion CNN: directly learns graph representations |
| GK | Graph Kernel Methods: uses sequence of subgraphs to capture features |
| LR | Logistic Regression: benchmark regression using 15 "standard" features |

All these methods are tested in a standard supervised learning framework where $n$ training examples $(x_i, y_i)$ are given (the input $x_i$ and $y_i$ the target label). For DBN, CNN and SdA, the input $x_i$ is our image representation of the subgraph. For DCNN and GK, the input $x_i$ is the graph itself, represented as an adjacency matrix. For LR, the input $x_i$ is a set of 15 classical features (assortativity, clustering coefficient, etc.).

### 2.3 CAFFE-BASED IMAGE CLASSIFICATION USING TRANSFER LEARNING

Caffe is a deep learning framework developed by Jia et al. (2014), Berkeley AI Research and by community contributors with expressive and modular architecture in mind. It has been extensively used in image classification and filter visualization, learning handwritten digital data, and style recognition among other things as seen in Benanne (2014). We use a pre-trained model that is trained on a crowd-sourced labeled data set ImageNet. As of 2016, ImageNet had more than 10 million hand-annotated images. The massive volume combined with a deep convolutional neural network gives us fine-grained discriminatory power for images.

Given an image, the output of our Caffe-based image classification function is a vector of (label, label-probability) tuples, sorted in decreasing order of probabilities. An example of the output for a real image is shown in Figure 2a. Although Caffe has not been trained on image embeddings of

graphs, such images nonetheless produce vectors that have sufficient discriminatory information that we extract using the post processing step (see Section 2.3.1). An example of a vector corresponding to a Facebook network sample is shown in Figure 2b.

Caffe provides either *maximally accurate* or *maximally specific* classification. We use the *maximally specific* categorization option in Caffe for our work. Further, while we have shown cardinality-5 vectors for brevity in Figure 2, we use cardinality-10 vectors in our experiments.

### 2.3.1 POST-PROCESSING FOR GRAPH CLASSIFICATION

Caffe provides a set of label vectors $L_i$ for each training network $x_i$. Each label vector is a tuple of (label, label-probability pairs) as deemed by Caffe. In this work, we ignore the probabilities, and treat each vector as an unordered list of labels (strings). Each training vector also has a *ground truth* parent label.

We use Jaccard similarity to compute a similarity metric between two label-vectors $L_j$ and $L_k$:

$$d(L_j, L_k) = \frac{|L_j \cap L_k|}{|L_j \cup L_k|}$$

We leave for future work the use of more sophisticated metrics which could use the probabilities from Caffe - our goal is to demonstrate the potential of even this simplest possible approach.

For a test graph, we get the label-vector $T$ from Caffe and then compute the $k$ nearest training vectors using the Jaccard distance $d(L_i, T)$ to each training vector $L_i$ and classify using the majority class $C$ among these $k$ nearest training vectors (ties are broken randomly). The test example is correctly classified if and only if its ground truth matches $C$. One advantage of the $k$ nearest neighbor approach is that it seamlessly extends to an arbitrary number of parent network classes.

### 2.4 DATASETS

We used a variety of datasets ranging from citation networks to social networks to e-commerce networks (see Table 1 and the brief descriptions in the Appendix). Our deep networks learn signatures from the image representation of each graph class. To gein some insight into these network signatures, we show the top principal component of the images of each network. This process is very similar to the one carried out in Turk & Pentland (1991). We grouped all the samples into their respective (9) categories. Then, we vectorized each sample; i.e., we reshaped each sample from $n \times n$ to $1 \times n^2$. We then performed principal component analysis on the vectorized dataset. We show the top principal component for each network in Figure 5a.

In Figure 5b, we show the structured image representations of sample 64-node subgraphs from each of the 9 datasets. These are adjacency matrices that have gone through the structuring process described in Section 2.1. Observe that the images for different graphs are well structured and quite different. This is why the deep networks are able to perform well at classifying the subgraphs, as we will see later in Section 3. Our approach leverages the structure as well as the distinctness of the image representations.

| Dataset | # Nodes | # Edges | Reference(s) |
|---|---|---|---|
| Citation | 34,546 | 421,578 | Leskovec et al. (2005); Gehrke et al. (2003) |
| Facebook | 4039 | 88,234 | Leskovec & Mcauley (2012) |
| Road Network | 1,088,092 | 1,541,898 | Leskovec et al. (2009) |
| Web | 875,713 | 5,105,039 | Leskovec et al. (2009) |
| Wikipedia | 4,604 | 119,882 | West & Leskovec (2012); West et al. (2009) |
| Amazon | 334,863 | 925,872 | Leskovec et al. (2007) |
| DBLP | 317,080 | 1,049,866 | Yang & Leskovec (2012) |
| Terrorist Net. | 271 | 756 | JJATT (2009) |
| Gowalla | 196,591 | 950,327 | Cho et al. (2011) |

Table 1: Datasets

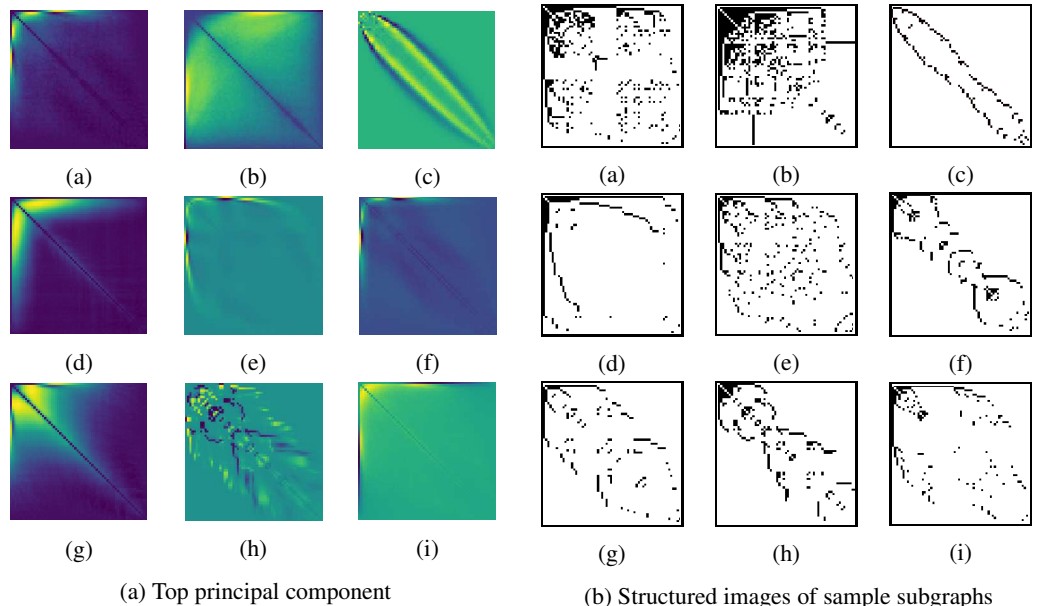

(a) Top principal component        (b) Structured images of sample subgraphs

Figure 5: Structured images and the top principal component. (a) Citation; (b) Facebook; (c) Road Network; (d) Web; (e) Wikipedia; (f) Amazon; (g) DBLP; (h) Terrorist Network; (i) Gowalla

# 3 EXPERIMENTAL SETUP AND RESULTS

In this section we will describe our experimental setup and present the results we obtained from the two approaches we have described in the earlier sections.

## 3.1 SUPERVISED CLASSIFICATION USING DEEP IMAGE CLASSIFIERS

We perform the graph classification task using the above mentioned 9 parent networks. We perform a random walk on each of these networks $5,000$ times and we stop when we get the required number of nodes per sample denoted by $n$. We carry out this exercise 4 times and set $n$ to 8, 16, 32 and 64 respectively. So, with 9 networks and $5,000$ samples per network, we create 4 datasets with $45,000$ samples each. Each dataset is of the size $45,000 \times n \times n$.

For a given dataset, we randomly chose $33\%$ of the dataset for validation and set aside $33\%$ of the dataset for testing. The accuracy score is defined as the ratio of the sum of the principal diagonal entries of the confusion matrix over the sum of all the entries of the confusion matrix (sometimes called the error matrix Stehman (1997)). A confusion matrix $C$ is such that $C_{i,j}$ is equal to the number of observations known to be in class $i$ but predicted (*confused*) to be in class $j$. We report the best accuracy score for each classifier in the following table.

|      | CNN  | SdA  | DBN  | DCNN | GK   | LR   |
|------|------|------|------|------|------|------|
| Acc. | 0.89 | 0.84 | 0.83 | 0.48 | 0.73 | 0.83 |

CNN was the best performing classifier while DCNN and GK were the poorest performers. This shows that off-the-shelf deep learning with graph image features with no tuning of the classifier is better than any other method we tested. Figure 6 summarizes the performance of all the methods we tested concisely. As expected, we obtained higher accuracy in classification as the number of nodes per sample $n$ increased. Note that straight line in the figure refers to the accuracy achieved for random guessing of the classes. We would like to point out the fact that even with only 8 nodes, we were able to do significantly better than random while being better than graph kernel methods and the feature based logistic regression classifier.

We would like to note that although LR performs okay, it is very hard to choose the features when graphs come from different domains. One set of features that worked best in one scenario may

not be the best in another. So, when using an untuned CNN that requires very little effort from the user handily outperforms these cumbersome methods, it is an easy choice to make. This is one of the biggest observations of our study. We also mixed the samples with different $n$'s to create hybrid datasets. We observed that the performance of the classification was better when the mixture had more samples with higher $n$. This is inline with our expectation and the results in Figure 6. Interested readers can refer to the Appendix for more details.

Graph kernel and feature-based methods performed better than DCNN, but not as well as the image embedding based methods. Kernel methods are complex and usually slow and the fact that they have performed poorly do not make them attractive. One may notice that the accuracy scores for LR are very comparable to SdA. However, we would like to remark that LR is a *lossy* method since it approximates the graph and boils it down to a handful of features. It is very hard to decide which features must be used and the choice may vary for graphs in different domains in order to get optimal results. However, our approach is completely *lossless*. The structured image representation of the graph has every bit of the information the adjacency matrix does. So, we would not have to make any compromises to get the best out of the data.

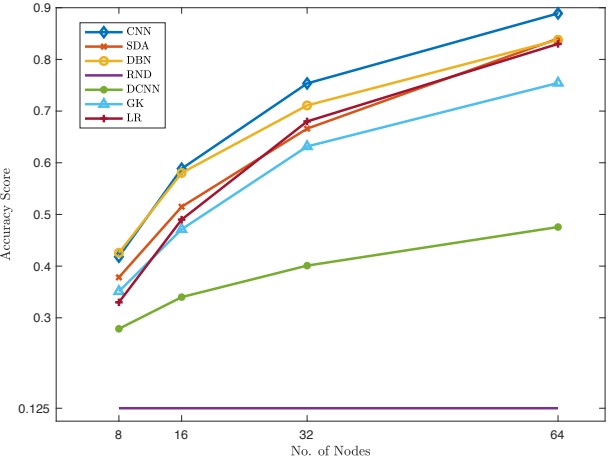

Figure 6: Classification accuracy increases as $n$ increases

We would like to make a note about DCNN. Out of the 4 neural network classification models we have used in this work, DCNN is the only one that takes a graph as an input instead of images. In fact, it takes two inputs: an adjacency matrix and a *design* matrix. The design matrix contains information about each node in the adjacency matrix. For example, information like average degree, clustering co-efficient etc. can be provided in the design matrix. In order to make the comparison between DCNN and the other classification models as *fair* as possible, we specified the values of the pixels in the image embedding in our design matrices throughout our experiments. When other information (assortativity, centrality etc.) were provided we observed no significant increase in performance. This is because these properties can be calculated from the graph which is already an input. The neural network is expected to have *learned* these features already.

## 3.2 Unsupervised Classification using Transfer Learning

In this section we present our experimental results of our second approach to graph classification: transfer learning. We show that our transfer learning approach is highly resilient to sparse training data. We achieve a respectable accuracy even when only 10% of the data was used for training.

Caffe can be treated as a black-box that requires very little interference from the user. This is significant because when one does not have access to ample data to train their own neural networks, transfer learning can be a very quick and effective fix to get the job done. These are the other two big observations to take home from our work.

We also carried out the following experiments: (a) differentiating networks of similar theme/type (i. Terrorist Net. vs Facebook, ii. Citation vs DBLP and iii. Web vs Wiki); and (b) multi-way

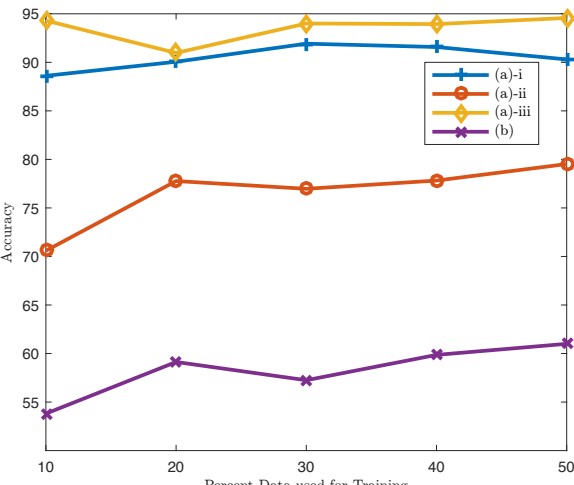

Figure 7: Accuracy of transfer learning. (a)-i: Terrorist Net. vs Facebook, (a)-ii: Citation vs DBLP and (a)-iii: Web vs Wiki and (b): multi-way classification.

classification. The detailed results are relegated to the Appendix, but we note that accuracy scores were in the high 90s and 80s for (a) and around the 60s for (b). While the multi-class scores are not as glamorous as those presented in Section 3.1, they are still worth mentioning.

Figure 7 plots the accuracy numbers from the (a) set of experiments as we progressively increase the proportion of data used for training. Each point on the $x$-axis shows the percentage of available data that was used for training, the reminder used for testing. Note that the reduction of training percentage hardly impacts accuracy except for a slight dip when only 10% is used for training. Most learning techniques, especially deep neural networks are sensitive to training data volume. The relative insensitivity of our approach is likely due to the fact that we leverage *pre-trained* recognition engine in the image domain which has already been trained with a massive volume of images. This shows that the transfer learning approach is very robust and resilient to sparse training data.

Finally, we study the impact of $k$, the neighborhood size for the majority rule, on accuracy. We show the detailed analysis in the Appendix but the take away fact is that as long as $k > 15$, we do not have to worry too much about tuning $k$, showing once again that the approach is robust.

## 4 CONCLUSION AND FUTURE WORK

Our experiments overwhelmingly show that the structured image representation of graphs achieves successful graph classification with ease. The image representation is *lossless*, that is the image embeddings contain all the information in the corresponding adjacency matrix. Our results also show that even with very little information about the parent network, Deep network models are able to extract network signatures. Specifically, with just 64-node samples from networks with up to 1 million nodes, we were able to predict the parent network with $> 90\%$ accuracy while being significantly better than random with only 8-node samples. Further, we demonstrated that the image embedding approach provides many advantages over graph kernel and feature-based methods.

We also presented an approach to graph classification using transfer learning from a completely different domain. Our approach converts graphs into 2D image embeddings and uses a pre-trained image classifier (Caffe) to obtain label-vectors. In a range of experiments with real-world data sets, we have obtained accuracies from 70% to 94% for 2-way classification and 61% for multi-way classification. Further, our approach is highly resilient to training-to-test ratio, that is, can work with sparse training samples. Our results show that such an approach is very promising, especially for applications where training data is not readily available (e.g. terrorist networks).

Future work includes improvements to the transfer learning by improving the distance function between label-vectors, as well as using the probabilities from Caffe. Further, we would also look

to generalize this approach to other domains, for example classifying radio frequency map samples using transfer learning.

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

# A  APPENDIX

## A.1  CLASSIFIERS FOR EXTRACTING NETWORK SIGNATURES

**Deep Belief Network (DBN)**  DBNs (see: Hinton (2009)) consist of multiple layers of unsupervised Restricted Boltzmann Machines (RBMs) where the output of each RBM is used as input to the next. (Hinton et al. (2006)). The RBMs can be trained greedily and a supervised back-propagation step can be used for fine-tuning.

Typically, DBNs have an input layer, hidden layer(s) and a final output layer. The input layer contains a node for each of the entries in the feature vector. For example, if the input is an image of size $8 \times 8$, then there will be $8 \times 8 = 64$ nodes in the input layer. The hidden layers consist of RBMs where the output of each of RBMs are used as input to the next. Finally, the output layer contains a node for each class. The probabilities of each class label is returned. The one with the highest probability is chosen as the overall classification for the given input.

**Convolutional Neural Network (CNN)**  CNNs are a category of neural networks that have been proven to be very effective in image classification tasks. Although they have been extensively used on real world images, we believe it is essential to include CNNs in our experiments.

The building blocks of a CNN are convolution layers, non-linear layers such as Rectified Linear Units (ReLU), pooling layers and fully connected layers for classification.

**Stacked De-Noising Auto-Encoder (SdA)**  We implement the stacked de-noising auto-encoder (SdA) based on greedy training algorithm presented in Wu & Magdon-Ismail (2016). In a regular multi-layer deep neural network, each layer is trained to "reconstruct" the input from the previous layer. Then, the system is fine tuned by using back-propagation. In SdA, instead of the original input, a noisy input is fed to the system.

**Diffusion-Convolutional Neural Network (DCNN)** This model, introduced by Atwood & Towsley (2016), works on the graphs themselves rather than the image embeddings of their adjacency matrices. DCNNs provide a flexible representation of graphical data that encodes node features, edge features, and purely structural information with little preprocessing. DCNNs learn diffusion-based representations from graph-structured data which is made possible by the new diffusion-convolution operation.

**Graph Kernel Approach** We use a graph kernel introduced in Shervashidze et al. (2011) which uses a rapid feature extraction scheme based on the Weisfeiler-Lehman test of isomorphism on graphs. It maps the original graph to a sequence of graphs, whose node attributes capture topological and label information. We use the code provided here Ghisu as-is in our experiments.

**Feature-based Classification** We compute 15 classic features for every graph sample and use just these to perform classification using logistic regression. The said features are: transitivity, average clustering co-efficient, average node connectivity, edge connectivity, average eccentricity, diameter, average shortest path, average degree, fraction of single-degree nodes, average closeness centrality, central points, density, average neighbor degree and top two eigen values of the adjacency matrix.

## A.2 Datasets

**Citation** This citation network is from Arxiv HEP-PH (high energy physics phenomenology). If a paper $i$ cites paper $j$, the graph contains a directed edge from $i$ to $j$. There are $34,546$ nodes with $421,578$ edges. See: Leskovec et al. (2005); Gehrke et al. (2003)

**Facebook** This social network contains "friends lists" from Facebook. There are $4039$ people (nodes) and there is an undirected edge between nodes if they are friends. There are $88,234$ such edges. See: Leskovec & Mcauley (2012)

**Road Network** This is a road network of Pennsylvania. Intersections and endpoints are represented by nodes, and the roads connecting these intersections are represented by undirected edges. There are $1,088,092$ nodes and $1,541,898$ edges in this network. See: Leskovec et al. (2009)

**Web** Nodes in this network represent web pages and directed edges represent hyperlinks between them. There are $875,713$ nodes and $5,105,039$ edges. See: Leskovec et al. (2009)

**Wikipedia** This is a Wikipedia hyperlink graph. A condensed version of Wikipedia was used in the collection of this dataset. There are $4,604$ articles (nodes) with $119,882$ links (edges) between them. See: West & Leskovec (2012); West et al. (2009)

**Amazon** This is a product co-purchase network of amazon.com. The nodes are products sold on amazon.com. There is an undirected edge between two products if they are frequently co-purchased. There are $334,863$ nodes and $925,872$ edges. See: Leskovec et al. (2007)

**DBLP** This is a co-authorship network. It has authors for its nodes and there is an undirected edge between them if they have co-authored at least one paper. There are $317,080$ nodes and $1,049,866$ edges. See: Yang & Leskovec (2012)

**Terrorist Network** We use the "Al Qaeda Operations Attack Series 1993-2003, Worldwide" dataset consisting of 271 nodes (participants of Al Qaeda terrorist group) with $756$ links between them. See: JJATT (2009)

**Gowalla** Gowalla was a location-based social networking website where users shared their locations by checking-in. The friendship network is undirected and consists of $196,591$ nodes and $950,327$ edges. See: Cho et al. (2011)

## A.3 Hybrid Datasets

We present the detailed results for different mixtures of datasets that we experimented with in the supervised setting. Although the performance deteriorates when different $n$'s are mixed, the relative ordering of the methods w.r.t. their performances remains the same. Table below shows the 4 cases and as Figure 8 shows, classification accuracy increases as the contribution by samples with higher $n$ increases.

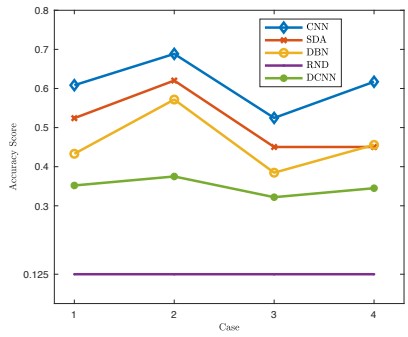

Figure 8: Performance on hybrid datasets

| Nodes | Case 1 | Case 2 | Case 3 | Case 4 |
|-------|--------|--------|--------|--------|
| 8 | 25% | 10% | 40% | 10% |
| 16 | 25% | 20% | 30% | 40% |
| 32 | 25% | 30% | 20% | 40% |
| 64 | 25% | 40% | 10% | 10% |

### A.4 TRANSFER LEARNING RESULTS CONTD.

We present the results from two sets of experiments: (a) telling apart networks of similar theme/type; and (b) multi-way classification. Tables 9a, 9b, and 9c show the results of classification between similarly-themed networks. We see that our approach is able to classify Wiki and Web with a very high accuracy of 95%. The other two are respectable as well, with 90% and 80% accuracies.

| Data | Prc. | Rec. | F1 |
|------|------|------|-----|
| Ter. Net. | 0.84 | 0.99 | 0.91 |
| FB | 0.99 | 0.81 | 0.89 |
| **Acc.** | | **90.3%** | |

(a) Terrorist Net. vs Facebook

| Data | Prc. | Rec. | F1 |
|------|------|------|-----|
| Cit. | 0.90 | 0.67 | 0.76 |
| DBLP | 0.74 | 0.92 | 0.82 |
| **Acc.** | | **79.51%** | |

(b) Citations vs DBLP

| Data | Prc. | Rec. | F1 |
|------|------|------|-----|
| Wiki | 0.96 | 0.93 | 0.94 |
| Web | 0.93 | 0.97 | 0.94 |
| **Acc.** | | **94.57%** | |

(c) Wiki vs Web

Figure 9: Classification between similarly themed networks

Table below shows that our approach does not do quite as well when it comes to multi-way classification. While multi-way classification is challenging in general, our particular approach of using the majority rule may be somewhat more impacted since now the correct class has to outnumber several other classes.

| Data | Prc. | Rec. | F1 |
|------|------|------|-----|
| Wiki | 0.71 | 0.79 | 0.75 |
| Web | 0.66 | 0.73 | 0.69 |
| DBLP | 0.57 | 0.59 | 0.58 |
| Terrorist Net. | 0.48 | 0.70 | 0.57 |
| Citations | 0.36 | 0.18 | 0.24 |
| Facebook | 0.84 | 0.66 | 0.74 |
| **Accuracy** | | **61%** | |

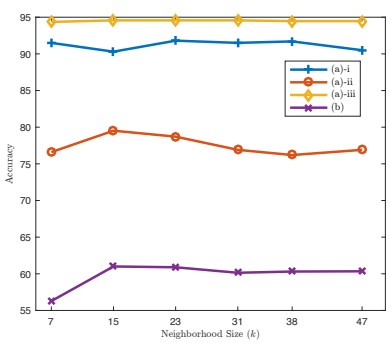

Figure 10: Accuracy vs nbhood. size $(k)$

Figure 10 shows the variation of accuracy results when $k$ is varied in steps of 8 from $k = 7$ to $k = 47$, with the base case $k = 15$ used for the tabulated results above included for comparison. As can be seen, except for a couple of cases with $k = 7$ providing a somewhat lower accuracy, the variation is within 1% of the base case. Thus, as long as $k > 15$, we do not have to worry too much about tuning $k$, showing that the approach is robust.

