# OpenReview forum: "Network Signatures from Image Representation of Adjacency Matrices: Deep/Transfer Learning for Subgraph Classification"
_ICLR.cc/2018/Conference — Reject_

### Official Review · AnonReviewer1 · 2017-11-17
**Surprising that the method works, but the method is too unprincipled for me to really see the value of it.**

**Rating:** 3
**Confidence:** 3

**Review:**

The paper proposes to use 2-d image representation techniques as a means of learning representations of graphs via their adjacency matrices. The adjacency matrix (or a subgraph of it) is first re-ordered to produce some canonical ordering which can then be fed into an image representation method. This can then be fed into a classifier.

This is a little too unprincipled for my taste. In particular the paper uses a Caffe reference model on top of the adjacency matrix, rather than learning a method specifically for graphs. Perhaps this is due to a lack of available graph training data, but it doesn't seem to make a lot of sense.

Maybe I missed or overlooked some detail, but I didn't spot exactly what the classification task was. I think the goal is to identify which of the graphs a subgraph belongs to? I'm not sure how relevant this graph classification task is.

The method does prove that the Caffe reference model maintains some information that can be used for classification, but this doesn't really suggest a generalizable method that we could confidently use for a variety of tasks. It's surprising that it works at all, but ultimately doesn't reveal a big scientific finding that could be re-used.

---

> ### Public Comment · (anonymous) · 2017-12-28
> **Response**
>
> Thank you for your comments. We address the main comments separately.
>
> 1. "It is surprising that the method works and it is unprincipled." The whole purpose of input representation is to find the right input representation such that a SIMPLE model with access to little data can perform good test prediction. It is not surprising that the method works since the encoding of the graph into the image is LOSSLESS. It is surprising that the graph image-feature works with simple models, that is the whole point of the paper. We do not understand in what sense the feature is unprincipled. The feature suitably reorders the vertices in the graph so that structural information information of the graph can be represented by spatial organization within the image. In the same vein, one could say that convolutional features which represent image information at different scales are unprincipled.
>
> 2. "but this doesn't really suggest a generalizable method." We showed results of blindly applying the feature to 9 different networks drawn from various application domains ranging from social networks to physical networks like road networks. The performance of this feature consistently outperforms all  other methods, which suggests that the method is generalizable.
>
> 3. "I didn't spot exactly what the classification task was" We tried to make this clear in Figure 1: Given a small subgraph, identify what type of parent graph it came from.
>
> 4. "I'm not sure how relevant this graph classification task is." Graph and subgraph classification is a large domain with a large amount of research ranging from standard learning methods based on classical graph features to kernel methods and so on. The specific task considered in this paper is merely a formalization of this task in which the class of a graph is determined by its parent graph.
>
> The intuition behind this work: Deep learning models are highly capable of classifying structured real world images. We leverage this strength to classify subgraphs using the structured image embeddings we obtain that are a lossless representation of graphs. We also show that even when using a model (Caffe) that was trained in a completely different domain (real world images - ImageNet), the structured representation is powerful enough to provide more than meaningful results. You mention "... the paper uses a Caffe reference model on top of the adjacency matrix, rather than learning a method specifically for graphs ..." in your review. We think that you are conflating the two very different approaches we are proposing in this paper.
>
> 5. "In particular the paper uses a Caffe reference model on top of the adjacency matrix, rather than learning a method specifically for graphs." We believe the reviewer missed this part of the paper. We demonstrated the valuused the image feature in TWO ways:
>
>   (a) (The main way) To train a classifier to classify graphs from scratch in a standard machine learning framework. Here we used several different learning models, including deep networks, kernels and standard models like regression based on classical graph features. Performance of our image feature is impressive (our opinion) compared to all other traditional features, and deep networks performed best.
>
>   (b) (The secondary way) In a pure transfer learning setting, we simply used the Caffe classifier trained on image-Net with NO further training except for applying a k-NN on the Caffe output. This mode of classification shows that the image can be treated as a traditional image since Caffe is trained on traditional images, and further, the performance is impressive, which means that this transfer setting can be used when there is limited training data in the original graph domain.
>
> We hope that we have addressed the concerns of the reviewer.

---

### Official Review · AnonReviewer3 · 2017-11-27
**Good paper**

**Rating:** 6
**Confidence:** 3

**Review:**

The paper proposed a subgraph image representation and validate it in image classification and transfer learning problems. The image presentation is a minor extension based on a method of producing permutation-invariant adjacency matrix. The experimental results supports the claim.

It is very positive that the figures are very helpful for delivering the information.

The work seems to be a little bit incremental. The proposed image representation is mainly based on a previous work of permutation-invariant adjacency matrix. A novelty of this work seems to be transforming a graph into an image. By the proposed representation, the authors are able to apply image classification methods (supervised or unsupervised) to subgraph classification.

It will be better if the authors could provide more details in the methodology or framework section.

The experiments on 9 networks support the claims that the image embedding approaches with their image representation of the subgraph outperform the graph kernel and classical features based methods. It seem to be promising when using transfer learning.

The last two process figures in 1.1 can be improved. No caption or figure number is provided.

It will be better to make the notations easy to understand and avoid any notation in a sentence without explanation nearby.
For example:
"the test example is correctly classified if and only if its ground truth matches C."(P5)
"We carry out this exercise 4 times and set n to 8, 16, 32 and 64 respectively."(P6)

Some minor issues:
"Zhu et al.(2011) discuss heterogeneous transfer learning where in they use..."(P3)
"Each label vector (a tuple of label, label-probability pairs)." (incomplete sentence?P5)

---

> ### Public Comment · (anonymous) · 2017-12-28
> **Response**
>
> Thank you for your review.
>
> 1. We have moved the extra details about datasets and classifiers from the methodology section to the Appendix in the interest of space. Is there any specific detail the reviewer expects to be included in this section?
>
> 2. We have fixed the other issues you have raised.

---

### Official Review · AnonReviewer2 · 2017-11-29
**interesting idea**

**Rating:** 6
**Confidence:** 3

**Review:**

This paper views graph classification as image classification, and shows that the CNN model adapted from image net can be effectively adapted to the graph classification. The idea is interesting and the result looks promising, but I do not understand the intuition behind the success of analogizing graph with images.

Fundamentally, a convolutional filter stands for a operation within a small neighborhood on the image. However, it is unclear how it means for the graph representation. Is the neighborhood predefined? Are the graph nodes pre-ordered?

I am also curious with the effect of pre-trained model from ImageNet. Since the graph presentation does not use color channels,  pre-trained model is used different from what it was designed to. I would imagine the benefit of using ImageNet is just to bring a random, high-dimensional embedding.  In addition, I wonder whether it will help to fine-tune the model on the graph classification data. Could this submission show some fine-tune experiments?

---

> ### Public Comment · (anonymous) · 2017-12-28
> **Response**
>
> Thank you for your review.
> We emphasize that the main focus of the paper is to present the power of the image feature created by lossless "embedding" of the adjacency matrix as an image. We use this image feature in two ways:
> 1. To train a classifier from scratch in a standard machine learning framework. Here we used several different models, including deep networks. Performance of this feature is impressive compared to other traditional features, and deep networks performed best.
> 2. In a pure transfer learning setting we simply used the Caffe classifier trained on image-Net with NO further training except for applying a k-NN on the Caffe output. This mode of classification shows that the image can be treated as a traditional image and can be used when there is limited training data.

---

### Comment · AnonReviewer2 · 2017-11-25
**what does the image filter mean for adjacent matrices?**

This paper views graph classification as image classification, and shows that the CNN model adapted from image net can be effectively adapted to the graph classification. The idea is interesting and the result looks promising, but I have difficulty to understand the intuition behind the success of analogizing graph with images. More specifically,  I wonder

1. what is the physical meaning of CNN filters respond to the graph representation?

2. for images from ImageNet, each pixel is represented by 3 color channels (RGB). Will the adjacent matrices representation use such channels?

3. if we shuffle the order of graph nodes,  the rows/columns  in adjacent matrix will exchange. Will the image based classification result be the same?

---

> ### Public Comment · (anonymous) · 2017-11-25
> **Response**
>
> Thanks for the comment. Please see the responses below.
>
> 1. what is the physical meaning of CNN filters respond to the graph representation?
> - I'm not sure I understood your question correctly. I'm assuming you meant image embeddings by "graph representation". From CNN's perspective, the structured image embeddings are like any other images. The fact that these structured image embeddings were obtained from adjacency matrices has no effect on CNN.
>
> 2. for images from ImageNet, each pixel is represented by 3 color channels (RGB). Will the adjacent matrices representation use such channels?
>
> - Caffe (trained on ImageNet) takes in black & white/grayscale images as input as well. The structured image embeddings of the adjacency matrices do not have to be modified in any way.
>
> 3. if we shuffle the order of graph nodes,  the rows/columns  in adjacent matrix will exchange. Will the image based classification result be the same?
>
> - Yes. The shuffling of the order of the nodes in the adjacency matrices does not affect classification. This is because we apply a structuring process on the matrices before we obtain the image embeddings. This ensures that no matter the arrangement of the nodes, the image embedding produces the same structure for a given adjacency matrix. It's permutation invariant. See Section 2.1 for details.

---

### Author Response · Authors · 2018-01-03
**Update**

The latest submission has the following updates:

1. Added captions to Figures 3 and 4 in Section 1
2. Fixed typos as pointed out by reviewer3 in sections 2.3.1, 3.1 and 4
3. Reworded a few phrases in Sections 4 and A

---

### Decision · Program_Chairs · 2018-01-29
**ICLR 2018 Conference Acceptance Decision**

**Decision:**

Reject

**Comment:**

The main idea of the paper is to transform graph classification into image representation (via adjacency matrices). Two reviewers are positive, while one is negative. The concerns are novelty (as mentioned by R2), while the last reviewer thinks the method is too simple and unprincipled (here the AC agrees with authors that simple is not necessarily bad). Overall, none of the reviewers champions this paper. Due to many excellent submissions, unfortunately this paper cannot be accepted in present form.